# Transcriptomics Reveals Host-Dependent Differences of Polysaccharides Biosynthesis in *Cynomorium songaricum*

**DOI:** 10.3390/molecules27010044

**Published:** 2021-12-22

**Authors:** Jie Wang, Hongyan Su, Hongping Han, Wenshu Wang, Mingcong Li, Yubi Zhou, Yi Li, Mengfei Li

**Affiliations:** 1Qinghai Key Laboratory of Qinghai-Tibet Plateau Biological Resource, Northwest Institute of Plateau Biology, Chinese Academy of Sciences, Xining 810008, China; wangjie@nwipb.cas.cn (J.W.); limingcong@nwipb.cas.cn (M.L.); 2University of Chinese Academy of Sciences, Beijing 100049, China; 3State Key Laboratory of Aridland Crop Science, College of Life Science and Technology, Gansu Agricultural University, Lanzhou 730070, China; Shy922322@163.com (H.S.); lmf@gsau.edu.cn (M.L.); 4School of Chemistry and Chemical Engineering, Qinghai Normal University, Xining 810008, China; sunnyhhp@163.com; 5Alxa Forestry and Grassland Research Institute, Alxa 750306, China; amlyswws@163.com

**Keywords:** *Cynomorium songaricum*, polysaccharides biosynthesis, transcriptomics analysis, *Nitraria roborowskii*, *Nitraria sibirica*

## Abstract

*Cynomorium songaricum* is a root holoparasitic herb that is mainly hosted in the roots of *Nitraria roborowskii* and *Nitraria sibirica* distributed in the arid desert and saline-alkaline regions. The stem of *C. songaricum* is widely used as a traditional Chinese medicine and applied in anti-viral, anti-obesity and anti-diabetes, which largely rely on the bioactive components including: polysaccharides, flavonoids and triterpenes. Although the differences in growth characteristics of *C. songaricum* between *N. roborowskii* and *N. sibirica* have been reported, the difference of the two hosts on growth and polysaccharides biosynthesis in *C. songaricum* as well as regulation mechanism are not limited. Here, the physiological characteristics and transcriptome of *C. songaricum* host in *N. roborowskii* (CR) and *N. sibirica* (CS) were conducted. The results showed that the fresh weight, soluble sugar content and antioxidant capacity on a per stem basis exhibited a 3.3-, 3.0- and 2.1-fold increase in CR compared to CS. A total of 16,921 differentially expressed genes (DEGs) were observed in CR versus CS, with 2573 characterized genes, 1725 up-regulated and 848 down-regulated. Based on biological functions, 50 DEGs were associated with polysaccharides and starch metabolism as well as their transport. The expression levels of the selected 37 genes were validated by qRT-PCR and almost consistent with their Reads Per kb per Million values. These findings would provide useful references for improving the yield and quality of *C. songaricum*.

## 1. Introduction

*Cynomorium songaricum* Rupr. is a root holoparasitic herb that is mainly hosted in the roots of *Nitraria* L., and widely distributed in the arid desert and saline-alkaline regions in northwest of China including: Qinghai, Xinjiang, Inner Mongolia and Ningxia [1,2]. As a traditional Chinese medicine, the stem of *C. songaricum* is generally used to tonify kidney yang, replenish essence and blood and relax the bowels [3,4]. In recent years, the stem has also been applied in anti-viral, anti-oxidation, anti-obesity, anti-diabetes, anti-tumor and ameliorates Alzheimer’s disease [5,6,7,8,9,10], which largely rely on the bioactive components including: polysaccharides (mainly polymerized by glucose, mannose and galactose), flavonoids (e.g., catechin, epicatechin and rutin), triterpenes (e.g., ursolic acid, acetyl ursolic acid and malonyl ursolic acid hemiester) and liposoluble constituents (e.g., hexadecanoic acid, oleic acid and docosenoic acid) [5,11,12,13,14,15,16].

The genus *Nitraria* L. is a perennial shrub and always used as a vital ecological protection plant for windbreak and sand fixation [17]. It contains 11 species in the world and 6 of them are in China [18]. *C. songaricum* is found to mainly host in four species including: *N. roborowskii* Kom., *N. sibirica* Pall., *N. tangutorum* Bobr. and *N. sphaerocarpa* Maxim [19,20]. Except the *N. sphaerocarpa*, the other three species mainly distribute in Qinghai, China [21]. Extensive surveys on habitat have found that *N. roborowskii* prefers locating in the margin of desert, *N. sibirica* in the salinized sand and drought hillslope and *N. tangutorum* is a transitional ecotype between *N. roborowskii* and *N. sibirica* [22,23]. Previous investigations into the differences in growth characteristics between *N. roborowskii* and *N. sibirica* have demonstrated that the growth indexes (e.g., seed weight, fruit weight and seedling height) of *N. roborowskii* are greater than *N. sibirica* [24]; while the salt tolerance, seed-setting rate, contents of nutritional components and trace elements of *N. sibirica* are higher than *N. roborowskii* [25,26,27,28,29,30].

*C. songaricum* is currently an endangered species, in large part because of an indiscriminate uprooting of wild plants to meet the increasing commercial demand of the pharmaceutical industry. As a holoparasitic herb, *C. songaricum* totally depends on the *Nitraria* L., for nutrients and water during the whole growth and development cycle [31]. *C. songaricum* is widely used as a traditional Chinese medicine and several pharmacological activities are largely relied on polysaccharides [10,15]; moreover, the growth differences in *C. songaricum* host in the two *N. roborowskii* and *N. sibirica* have been reported [22,23,24], the regulation mechanism of polysaccharides biosynthesis has not been revealed. Thus, it is urgent and necessary to identify the optimization host to increase production of *C. songaricum*. Up to now, studies on the effect of different hosts on growth and metabolite accumulation of *C. songaricum* have not been conducted. This study examines biomass, soluble sugar accumulation, antioxidant capacity and transcriptional alternations of stem between CR and CS.

## 2. Results

### 2.1. Comparison of Growth Characteristics between CR and CS

As shown in Figure 1, significant differences in growth characteristics of stems between the CR and CS were observed, with FW of total stems, FW per stem, stem length and diameter of CR exhibiting a 5.1-, 3.3-, 1.4 and 1.3-fold increase compared to that of CS, respectively.

### 2.2. Comparison of Soluble Sugar Content and Antioxidant Capacity between CR and CS

As shown in Figure 2, significant differences in soluble sugar content and antioxidant capacity between the CR and CS were observed, with a 1.1-, 1.5- and 1.5-fold respective decrease of soluble sugar content, DPPH scavenging activity and FRAP value on an FW basis in stem of CR compared to that of CS (Figure 2A,C,E), while a 3.0-, 2.1- and 2.1-fold increase on a per stem basis (Figure 2B,D,F).

### 2.3. Global Gene Analysis

To reveal the differences of carbohydrate metabolism between the CR and CS, comparison of the transcripts were performed. A robust data was collected, 51.2 and 46.8 million high-quality reads were obtained after data filtering, and 42.5 and 39.5 million unique reads as well as 1.6 and 1.4 million multiple reads were mapped from the CR and CS, respectively (Figure 3; Appendix A). Total 95,126 unigenes were annotated on KEGG (10,274), KOG (17,550), Nr (40,427) and Swissprot (16,181) databases (Figure 4), and the top 10 species distribution against Nr includes: *Cajanus cajan*, *Vitis vinifera*, *Cephalotus follicularis*, *Theobroma cacao*, *Nicotiana attenuata*, *Juglans regia*, *Corchorus capsularis*, *Brassica napus*, *Brassica rapa* and *Medicago truncatula* (Figure 5).

A total of 16,921 DEGs were identified in the CR compared with CS, with 6580 genes up-regulated (UR) and 10,341 genes down-regulated (DR) (Figure 6). Of these 16,921 DEGs, 2684 genes were identified to match with the databases (Figure 7A). Among the 2684 genes, 2573 genes with known functions were partitioned into 1725 UR and 848 DR (Figure 7B,C).

### 2.4. Biological Category of DEGs

Based on biological functions, the 2573 genes were divided into nine categories: primary metabolism (493), transport (371), transcription factor (426), cell morphogenesis (289), bio-signaling (287), stress response (224), translation (195), secondary metabolism (179) and photosynthesis and energy (109) (Figure 7C; Appendix A). Based on carbohydrate metabolism driving genes characterized, 50 DEGs (32UR and 18DR) were identified as potential regulatory genes for polysaccharides and starch metabolism (37) as well as transport (13) (Figure 7C; Table 1).

### 2.5. DEGs Involved in Carbohydrate Metabolism and Transport

#### 2.5.1. DEGs Involved in Polysaccharides Metabolism

Thirty-two DEGs, presenting 21 UR and 11 DR in the CR compared with CS, directly participate in polysaccharides metabolism including: glucose (*GapA*, *GAPA1*, *GAPA2*, *GAPB*, *GAPC*, *PGMP*, and *UGP1*), galactose (*BGAL*, *BGAL5*, *BGAL7*, *GALM*, *GALT29A*, *GLCAT14A*, and *GOLS2*), mannose (*CYT1*, *GMD1*, *MAN5*, *MSR2 MUR1*, and *PMI2*), fucose (*OFUT9*, *OFUT20*, *OFUT23*, *OFUT27*, and *OFUT35*), trehalose (*TPS7*, *TPS9*, *TPS11*, *TPPF*, and *TPPJ*) and fructose (*CWINV1* and *CYFBP*) (Table 1). Here, 22 genes were selected to be validated by qRT-PCR, and their RELs were consistent with the RPKM values, with UR for metabolism of glucose, mannose, trehalose and fructose (Figure 8A–D), while differential expression for fucose metabolism (UR for the *OFUT9* and DR for the *OFUT20*, *OFUT23* and *OFUT27*) (Figure 8E), and DR for galactose metabolism (Figure 8F).

#### 2.5.2. DEGs Involved in Starch Metabolism

Five DEGs, presenting two UR and three DR in the CR compared with CS, directly participate in starch metabolism including: *At2g31390*, *DSP4*, *NANA*, *SBE2.2* and *SS2* (Table 1). These genes were validated by qRT-PCR, and their RELs were consistent with the RPKM values, with UR 3.5- and 6.8-fold for the *At2g31390* and *SS2*, and DR 0.6-, 0.9- and 0.6-fold for the *DSP4*, *NANA* and *SBE2.2*, respectively (Figure 9).

#### 2.5.3. DEGs Involved in Carbohydrate Transport

Thirteen DEGs, presenting nine UR and four DR in the CR compared with CS, are involved in carbohydrate transport including: *At1g67300*, *ERD6*, *MST1*, *STP1*, *STP5*, *STP12*, *STP13*, *SWEET5*, *SWEET12*, *SWEET14*, *SWEET15*, *UXT2* and *UXT3* (Table 1). Here, 10 genes were validated by qRT-PCR, and their RELs were consistent with the RPKM values, with the UR 4.5-, 7.3-, 4.6-, 3.5-, 4.5-, 6.2- and 1.5-fold for the *STP1*, *STP12*, *STP13*, *SWEET5*, *SWEET14*, *SWEET15* and *UXT2*, while the DR 0.5-, 0.1- and 0.8-fold for the *STP5*, *SWEET12* and *UXT3*, respectively (Figure 10).

## 3. Discussion

Although differences in growth characteristics and nutritional components of *C. songaricum* among the host species, especially in *N. roborowskii* and *N. sibirica*, have been observed in previous studies [25,26,27,28,29,30], the mechanism responsible for host-dependent growth and bioactive compound biosynthesis has not been dissected. Here, we found that there is a greater biomass, soluble sugar content and antioxidant capacity on a per stem basis in the CR than the CS (Figure 1 and Figure 2). By transcriptomics analysis in the CR compared with CS, a total of 2573 characterized genes differentially expressed with 1725 UR and 848 DR (Figure 7). By grouping genes based on biological functions, 50 genes (32 UR and 18 DR) were associated with carbohydrate metabolism and transport (Figure 7; Table 1).

Carbohydrates, one of the most abundant and widespread biomolecules in nature, not only plays an important role in plant growth and development, but also represents a treasure trove of untapped potential for pharmaceutical applications [32,33]. In this study, 37 genes were found to be involved in carbohydrate metabolism including polysaccharides (glucose, galactose, mannose, fucose, trehalose and fructose) and starch (Table 1). Among the 37 genes, 23 genes (62%) presenting up-regulated and 14 genes (38%) down-regulated suggest that the level of carbohydrate metabolism is greater in the CR than CS, which is in accordance with the higher content of soluble sugar on a per stem basis in the CR (Figure 2A,B).

For the polysaccharides metabolism, specifically, seven genes associated with glucose metabolic process include: *GapA*, *GAPA1*, *GAPA2*, *GAPB* and *GAPC* participating in the pathway Calvin cycle by catalyzing the reduction of 1,3-diphosphoglycerate by NADPH [34]; *PGMP* participating in both the breakdown and synthesis of glucose [35]; and *UGP1* converting glucose 1-phosphate to UDP-glucose and being essential for the synthesis of sucrose, starch, cell wall and callose deposition [36,37]. Seven genes associated with galactose metabolic process include:*BGAL*, *BGAL5* and *BGAL7* degrading polysaccharides by hydrolyzing terminal non-reducing beta-D-galactose residues in beta-D-galactosides [34]; *GALM* catalyzing the interconversion of beta-D-galactose and alpha-D-galactose [34]; *GALT29A* and *GLCAT14A* involved in the biosynthesis of type II arabinogalactan by, respectively, transferring galactose and glucuronate to oligosaccharides [38,39]; and *GOLS2* involved in the biosynthesis of raffinose family oligosaccharides [38]. Six genes associated with mannose metabolic process include: *CYT1* participating in synthesizing GDP-alpha-D-mannose from alpha-D-mannose 1-phosphate [40]; *GMD1* and *MUR1* catalyzing the conversion of GDP-D-mannose to GDP-4-dehydro-6-deoxy-D-mannose [41]; *MAN5* hydrolyzing the 1,4-beta-D-mannosidic linkages in mannans, galactomannans and glucomannans [42]; *MSR2* involved in mannan biosynthesis [43]; and *PMI2* involved in the synthesis of the GDP-mannose and dolichol-phosphate-mannose required for a number of critical mannosyl transfer reactions [44]. Five genes associated with fucose metabolic process include: *OFUT9*, *OFUT20*, *OFUT23*, *OFUT27* and *OFUT35* participating in the biosynthesis of matrix polysaccharides [45]. Five genes associated with trehalose metabolic process include: *TPS7*, *TPS9*, TPS11, *TPPF* and *TPPJ* involved in the trehalose biosynthesis [34,46]. Two genes associated with fructose metabolic process include: *CWINV1* hydrolyzing the terminal non-reducing beta-D-fructofuranoside residues in beta-D-fructofuranosides [47,48]; and *CYFBP* catalyzing fructose-1,6-bisphosphate to fructose-6-phosphate and inorganic phosphate [34,49].

For the starch metabolism, five genes associated with starch metabolic process include: *At2g31390* involved in maintaining the flux of carbon towards starch formation [34]; *DSP4* controlling the starch accumulation and acting as a major regulator of the initial steps of starch degradation at the granule surface [50]; *NANA* regulating endogenous sugar levels (e.g., sucrose, glucose and fructose) by modulating starch accumulation and remobilization [51]; *SBE2.2* involved in starch biosynthesis and catalyzing the formation of the alpha-1, 6-glucosidic linkages in starch [52]; and *SS2* participating in the pathway starch biosynthesis [53].

Transport plays critical roles in distribution and storage of carbohydrate from leaves to roots or other organs that required nutrition [54]. In this study, 13 genes were involved in carbohydrate transport with nine genes (69%) up-regulated and four genes (31%) down-regulated, suggesting that the ability of carbohydrate transport is stronger in the CR than the CS (Table 1). Specially, the 13 genes include: *At1g67300* participating in the efflux of glucose towards the cytosol [55]; *ERD6* participating in sugar transport [56]; *MST1* mediating active uptake of hexoses [57]; *STP1*, *STP5*, *STP12* and *STP13* participating in transporting glucose, 3-*O*-methylglucose, fructose, xylose, mannose, galactose, fucose, 2-deoxyglucose and arabinose [58]; SWEETs is a unique new family of sugar transporters that lead to many elusive transport steps including nectar secretion, phloem loading and post-phloem unloading as well as novel vacuolar transporters [59]. Here, four SWEETs genes *SWEET5*, *SWEET12*, *SWEET14* and *SWEET15* participate in phloem loading by mediating export from parenchyma cells feeding H^+^-coupled import into the sieve element/companion cell complex [59,60]; and *UXT2* and *UXT3* participate in transporting UDP-xylose and UMP [61].

## 4. Materials and Methods

### 4.1. Plant Materials

Stems of *C. songaricum* at vegetative growth stage, were host in the roots of *N. roborowskii* and *N. sibirica* (Figure 11) were collected on 6 May 2019 from Dulan county (2800 m; 36°2′25″ N, 97°40′26″ E) of Qinghai, China. The stems were cleaned and rapidly frozen in liquid nitrogen, the middle parts of stem were used for determination of soluble sugar content and antioxidant capacity, and the shoot apical meristems (SAM) were used for transcriptomic analysis.

### 4.2. Measurement of Growth Characteristics

Growth characteristics including fresh weight (FW) of total stems, FW per stem, and its length and diameter were immediately measured after the stems of *C. songaricum* were dug out and cleaned with running water and absorbent paper.

### 4.3. Determination of Soluble Sugar Content and Antioxidant Capacity

#### 4.3.1. Extracts Preparation

Fresh stems (1.0 g) were ground into homogenate by adding ethanol (20 mL), agitated at 120 r/min and 22 °C for 72 h, then centrifuged at 5000 r/min and 4 °C for 10 min. The supernatant was increased 20 mL with ethanol and then kept at 4 °C for measurement.

#### 4.3.2. Determination of Soluble Sugar Content

Soluble sugar content was determined by a phenol–sulfuric acid method [62,63]. Briefly, extracts (20 μL) were added in the reaction, absorbance reader was taken at 485 nm and soluble sugar content was calculated based on mg of sucrose.

#### 4.3.3. Determination of Antioxidant Capacity

Antioxidant capacity was determined by DPPH and FRAP methods [64,65]. DPPH radical scavenging assay was determined according to the description of Nencini et al. [66] and Li et al. [63]. Briefly, extracts (5 μL) were added in the reaction, absorbance reader was taken at 515 nm and the capacity to scavenge DPPH radicals was calculated as following Equation (1):DPPH scavenging activity (%) = [(*A*_0_ − *A*)/*A*_0_] × 100(1)
where “*A*_0_” and “*A*” were the absorbance of DPPH without and with sample, respectively.

FRAP assay was determined according to the description of Benzie and Strain [67]. Briefly, extracts (20 μL) were added in the reaction, absorbance reader was taken at 593 nm and the FRAP value was calculated on the basic of (FeSO_4_·7H_2_O, 500 μmol Fe (Ⅱ)/g) as following Equation (2):FRAP value (µmol Fe(II)/g) = [(*A* − *A*_0_)/(*A_FeSO4·7H2O_* − *A*_0_)] × 500 (µmol Fe(II)/g)(2)
where “*A*_0_” and “*A*” were the absorbance of FRAP without and with sample, respectively; *A_FeSO4·7H2O_* was the absorbance of FeSO_4_·7H_2_O.

### 4.4. Total RNA Extraction, Illumina Sequencing, Sequence Filtration, Assembly, Unigene Expression Analysis and Basic Annotation

Total RNA samples of CR and CS with three biological replicates were extracted using an RNA kit (R6827, Omega Bio-Tek, Inc., Norcross, GA, USA). The processes of enrichment, fragmentation, reverse transcription, synthesis of the second-strand cDNA and purification of cDNA fragments was applied following previous protocols [68]. RNA-seq was performed by an Illumina HiSeqTM 4000 platform (Gene Denovo Biotechnology Co., Ltd., Guangzhou, China). Raw reads were filtered according to previous descriptions [68]. Clean reads were assembled using Trinity [69]. The expression level of each transcript was normalized to RPKM [70], and DEGs were analyzed according to a criterion of |log_2_(fold-change)| ≥ 1 and *p* ≤ 0.05 by DESeq2 software and the edgeR package [71,72]. Unigenes were annotated against the databases including: NR, Swiss-Prot, KEGG, KOG and GO [73].

### 4.5. qRT-PCR Validation

The primer sequence (Table 2) was designed via a primer-blast in NCBI and synthesized by reverse transcription (Sangon Biotech Co., Ltd., Shanghai, China). First cDNA was synthesized using a RT Kit (KR116, Tiangen, China). PCR amplification was performed using a SuperReal PreMix (FP205, Tiangen, China). Melting curve was analyzed at 72 °C for 34 s. *Actin* gene was used as a reference control. The RELs of genes were calculated using a 2^−ΔΔCt^ method [74].

### 4.6. Statistical Analysis

All the measurements were performed using three biological replicates. A *t*-test was applied for independent samples, with *p* < 0.05 considered significant.

## 5. Conclusions

From the above observations, the stem biomass and polysaccharides accumulation of *C.*
*songaricum* host in *N. roborowskii* are significantly greater than that of *N. sibirica*. A total of 1725 UR and 848 DR genes were observed in CR compared to CS, and 50 DEGs were involved in polysaccharides biosynthesis, which indicates that the polysaccharides biosynthesis in *C.*
*songaricum* is host-dependent. The specific roles of candidate genes in regulating polysaccharides biosynthesis will require additional studies.

## Figures and Tables

**Figure 1 molecules-27-00044-f001:**
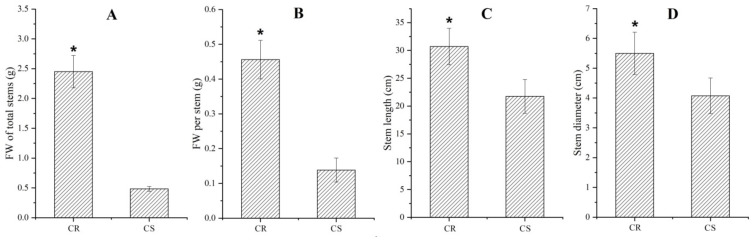
Growth characteristics of stems of *Cynomorium*
*songaricum* host in *Nitraria roborowskii* (CR) and *Cynomorium*
*songaricum* host in *Nitraria sibirica* (CS) (mean ± SD, n = 20). Images (**A**–**D**) represent FW of total stems, FW per stem, stem length and diameter, respectively. A t-test was applied for independent samples, the “*” is considered significant at *p* < 0.05 between CR and CS.

**Figure 2 molecules-27-00044-f002:**
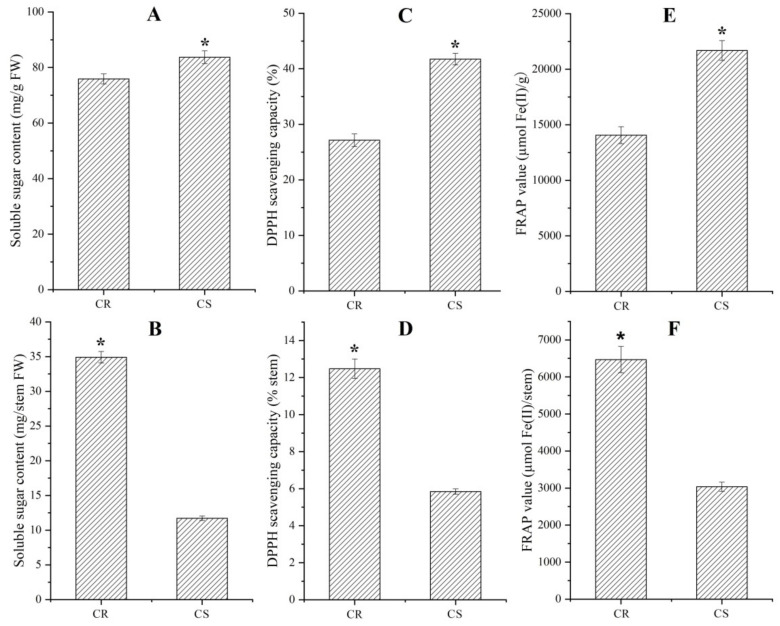
Soluble sugar content and antioxidant capacity in stems between the CR and CS (mean ± SD, n = 20). Images (**A**–**D**) as well as (**E**,**F**) represent soluble sugar content, DPPH scavenging activity as well as FRAP value on an FW and per stem basis, respectively. A t-test was applied for independent samples, the “*” is considered significant at *p* < 0.05 between CR and CS.

**Figure 3 molecules-27-00044-f003:**
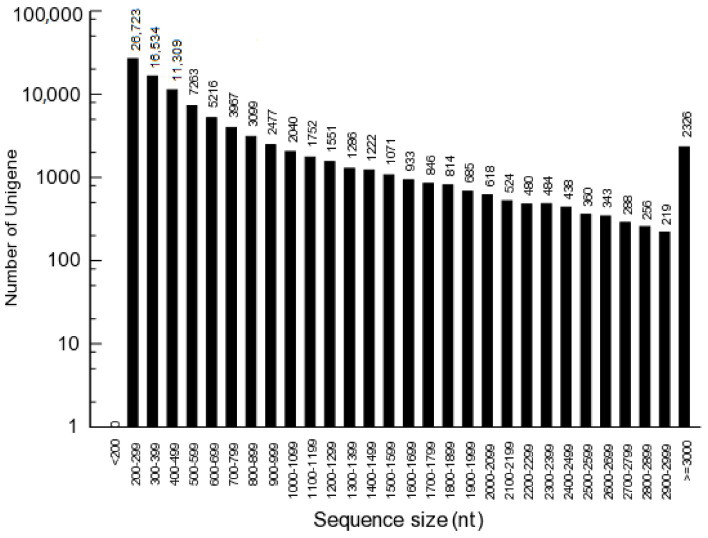
Length distribution of assembled unigenes in *C. songaricum*.

**Figure 4 molecules-27-00044-f004:**
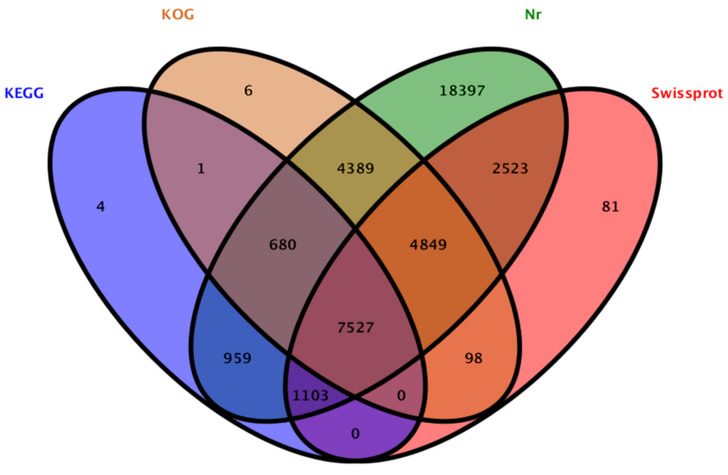
Basic annotation for all unigenes in *C. songaricum* on KEGG, KOG, Nr and Swissprot databases.

**Figure 5 molecules-27-00044-f005:**
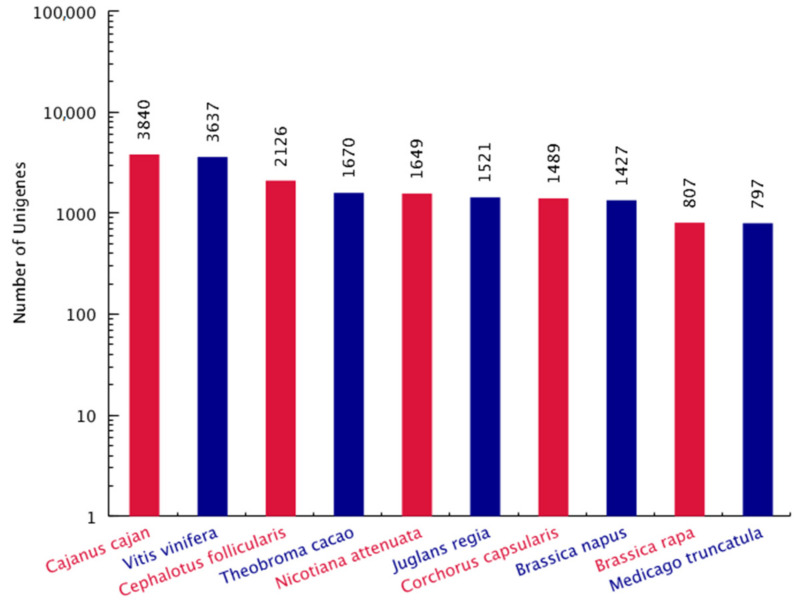
Top 10 species distribution of unigenes against Nr database.

**Figure 6 molecules-27-00044-f006:**
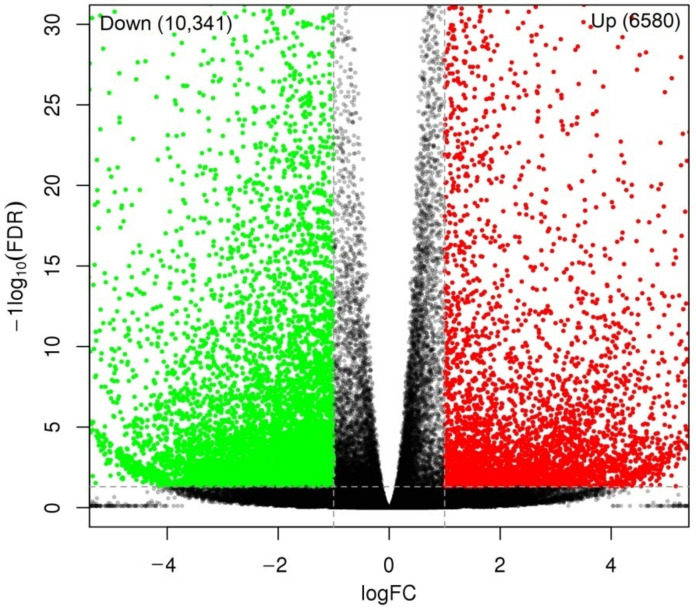
Volcano plot of unigenes and number of differentially expressed genes (DEGs) in the CR compared with CS.

**Figure 7 molecules-27-00044-f007:**
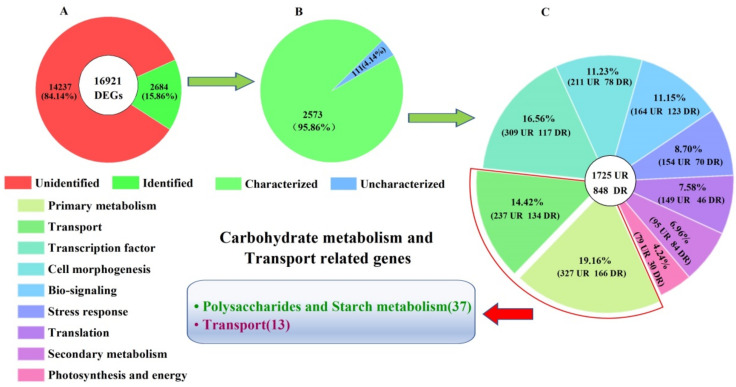
Distribution and classification of DEGs in the CR compared with CS (UR, up-regulation; DR, down-regulation). Image (**A**) represents the classification of unidentified and identified genes, image (**B**) represents the classification of uncharacterized and characterized genes and image (**C**) represents the classification of the functional genes.

**Figure 8 molecules-27-00044-f008:**
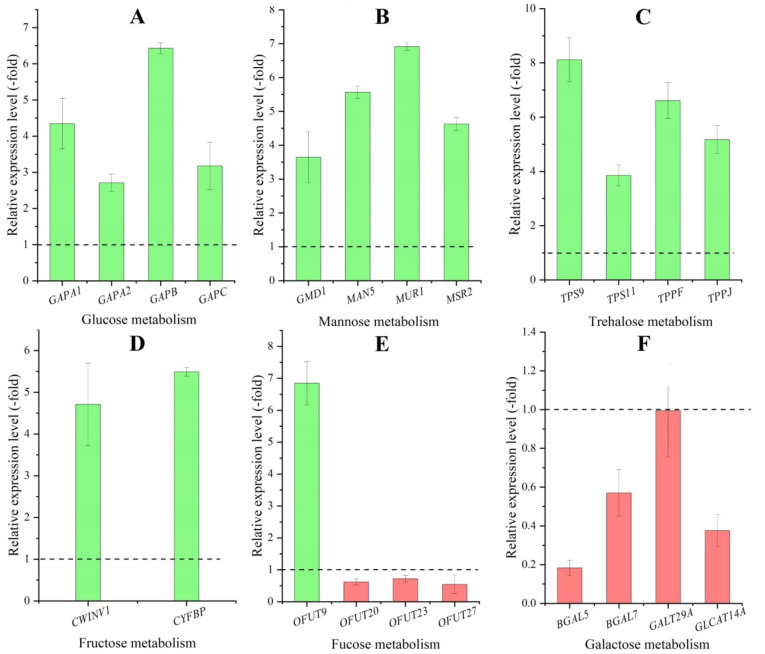
The relative expression level of genes involved in metabolism process of glucose (**A**), mannose (**B**), trehalose (**C**), fructose (**D**), fucose (**E**) and galactose (**F**) in the CR compared with CS, as determined by qRT-PCR. Column highlighted in green represents genes UR and red represents genes DR. The dotted line in the images differentiates UR (>1) and DR (<1) in CR compared with CS, represented. The same below.

**Figure 9 molecules-27-00044-f009:**
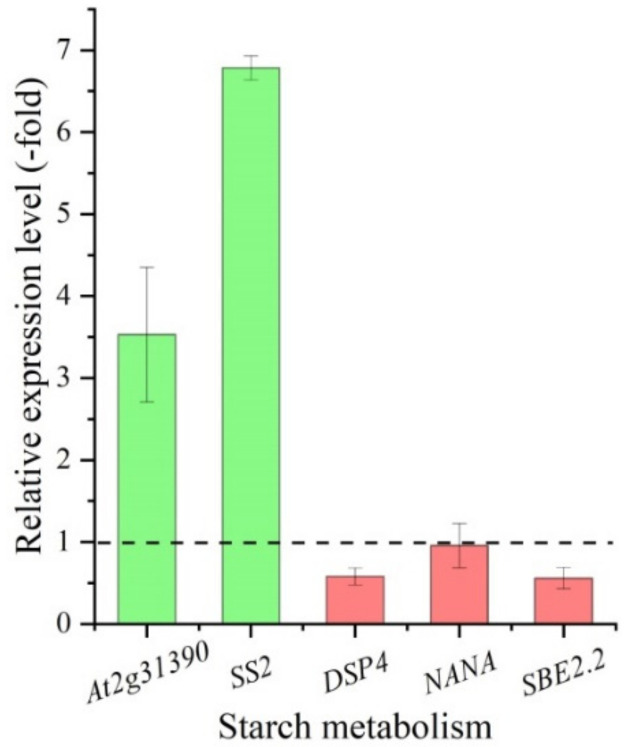
The relative expression level of genes involved in starch metabolism in the CR compared with CS, as determined by qRT-PCR.

**Figure 10 molecules-27-00044-f010:**
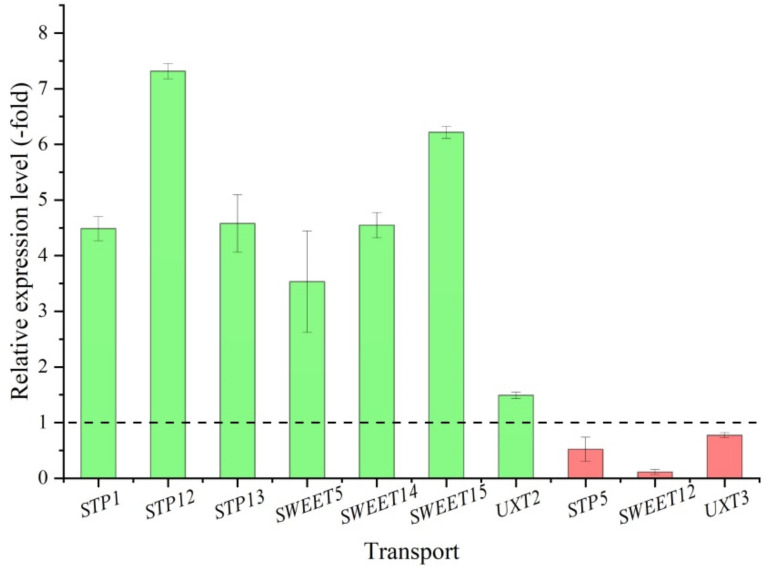
The relative expression level of genes involved in transport in the CR compared with CS, as determined by qRT-PCR.

**Figure 11 molecules-27-00044-f011:**
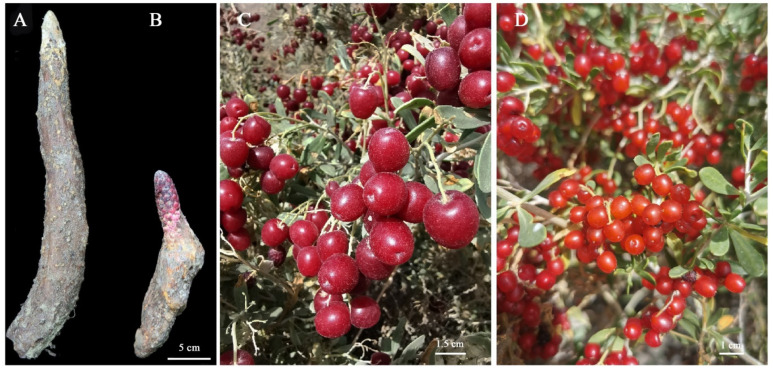
Morphological characteristics of stems of *C. songaricum* at vegetative growth stage and aerial parts of *N. roborowskii* and *N. sibirica*. Images (**A**,**B**) represent stems host in the roots of *N. roborowskii* and *N. sibirica*, and Images (**C**,**D**) represent aerial parts of *N. roborowskii* and *N. sibirica*, respectively.

**Table 1 molecules-27-00044-t001:** DEGs involved in carbohydrate metabolism and transport in the CR compared with CS.

Gene Name	Swissprot-ID	Protein Name	RPKM (CR/CS)
Polysaccharides Metabolism (32)				
Glucose (7)			
*GapA*	sp|Q8VXQ9|G3PA_COEVA	Glyceraldehyde-3-phosphate dehydrogenase A	8.83
*GAPA1*	sp|P25856|G3PA1_ARATH	Glyceraldehyde-3-phosphate dehydrogenase GAPA1	5.47
*GAPA2*	sp|Q9LPW0|G3PA2_ARATH	Glyceraldehyde-3-phosphate dehydrogenase GAPA2	4.37
*GAPB*	sp|P25857|G3PB_ARATH	Glyceraldehyde-3-phosphate dehydrogenase GAPB	7.25
*GAPC*	sp|P04796|G3PC_SINAL	Glyceraldehyde-3-phosphate dehydrogenase	3.25
*PGMP*	sp|Q9SM59|PGMP_PEA	Phosphoglucomutase	−1.70
*UGP1*	sp|P57751|UGPA1_ARATH	UTP-glucose-1-phosphate uridylyltransferase 1	2.80
**Galactose (7)**			
*BGAL*	sp|P48981|BGAL_MALDO	Beta-galactosidase	−1.00
*BGAL5*	sp|Q9MAJ7|BGAL5_ARATH	Beta-galactosidase 5	−1.32
*BGAL7*	sp|Q9SCV5|BGAL7_ARATH	Beta-galactosidase 7	−3.29
*GALM*	sp|Q5EA79|GALM_BOVIN	Aldose 1-epimerase	1.34
*GALT29A*	sp|Q9SGD2|GT29A_ARATH	Beta-1,6-galactosyltransferase GALT29A	−3.71
*GLCAT14A*	sp|Q9FLD7|GT14A_ARATH	Beta-glucuronosyltransferase GlcAT14A	−1.10
*GOLS2*	sp|C7G304|GOLS2_SOLLC	Galactinol synthase 2	−1.23
**Mannose (6)**			
*CYT1*	sp|O22287|GMPP1_ARATH	Mannose-1-phosphate guanylyltransferase 1	3.29
*GMD1*	sp|Q9SNY3|GMD1_ARATH	GDP-mannose 4,6 dehydratase 1	2.98
*MAN5*	sp|P93031|GMD2_ARATH	Mannan endo-1,4-beta-mannosidase 5	3.21
*MSR2*	sp|Q6YM50|MAN5_SOLLC	Protein MANNAN SYNTHESIS-RELATED 2	1.72
*MUR1*	sp|Q0WPA5|MSR2_ARATH	GDP-mannose 4,6 dehydratase 2	1.24
*PMI2*	sp|Q9FZH5|MPI2_ARATH	Mannose-6-phosphate isomerase 2	1.61
**Fucose (5)**			
*OFUT9*	sp|Q8H1E6|OFUT9_ARATH	O-fucosyltransferase 9	1.16
*OFUT20*	sp|O64884|OFT20_ARATH	O-fucosyltransferase 20	−2.52
*OFUT23*	sp|Q9MA87|OFT23_ARATH	O-fucosyltransferase 23	−1.86
*OFUT27*	sp|Q8GZ81|OFT27_ARATH	O-fucosyltransferase 27	−1.19
*OFUT35*	sp|Q94BY4|OFT35_ARATH	O-fucosyltransferase 35	1.14
**Trehalose (5)**			
*TPS7*	sp|Q9LMI0|TPS7_ARATH	Probable alpha,alpha-trehalose-phosphate synthase	−1.40
*TPS9*	sp|Q9LRA7|TPS9_ARATH	Probable alpha,alpha-trehalose-phosphate synthase	8.76
*TPS11*	sp|Q9ZV48|TPS11_ARATH	Probable alpha,alpha-trehalose-phosphate synthase	3.34
*TPPF*	sp|Q9SU39|TPPF_ARATH	Probable trehalose-phosphate phosphatase F	2.27
*TPPJ*	sp|Q5HZ05|TPPJ_ARATH	Probable trehalose-phosphate phosphatase J	2.48
**Fructose (2)**			
*CWINV1*	sp|Q43866|INV1_ARATH	Beta-fructofuranosidase, insoluble isoenzyme CWINV1	1.40
*CYFBP*	sp|Q9MA79|F16P2_ARATH	Fructose-1,6-bisphosphatase	2.29
**Starch Metabolism (5)**			
*At2g31390*	sp|Q9SID0|SCRK1_ARATH	Probable fructokinase-1	2.93
*DSP4*	sp|G4LTX4|DSP4_CASSA	Phosphoglucan phosphatase DSP4, amyloplastic	−1.95
*NANA*	sp|Q9LTW4|NANA_ARATH	Aspartic proteinase NANA	−3.64
*SBE2.2*	sp|Q9LZS3|GLGB2_ARATH	1,4-alpha-glucan-branching enzyme 2-2	−1.79
*SS2*	sp|Q43847|SSY2_SOLTU	Granule-bound starch synthase 2	4.05
**Carbohydrate Transport (13)**			
*At1g67300*	sp|Q9FYG3|PLST2_ARATH	Probable plastidic glucose transporter 2	1.12
*ERD6*	sp|O04036|ERD6_ARATH	Sugar transporter ERD6	2.47
*MST1*	sp|Q0JCR9|MST1_ORYSJ	Sugar transport protein MST1	−1.09
*STP1*	sp|P23586|STP1_ARATH	Sugar transport protein 1	8.84
*STP5*	sp|Q93Y91|STP5_ARATH	Sugar transport protein 5	−1.29
*STP12*	sp|O65413|STP12_ARATH	Sugar transport protein 12	5.61
*STP13*	sp|Q94AZ2|STP13_ARATH	Sugar transport protein 13	3.28
*SWEET5*	sp|Q9FM10|SWET5_ARATH	Bidirectional sugar transporter SWEET5	2.05
*SWEET12*	sp|O82587|SWT12_ARATH	Bidirectional sugar transporter SWEET12	−2.05
*SWEET14*	sp|Q2R3P9|SWT14_ORYSJ	Bidirectional sugar transporter SWEET14	1.57
*SWEET15*	sp|P0DKJ5|SWT15_VITVI	Bidirectional sugar transporter SWEET15	9.57
*UXT2*	sp|Q8GUJ1|UXT2_ARATH	UDP-xylose transporter 2	1.71
*UXT3*	sp|Q8RXL8|UXT3_ARATH	UDP-xylose transporter 3	−1.81

**Table 2 molecules-27-00044-t002:** Sequences of primer employed in qRT-PCR analysis.

Genes	Sequences (5′ to 3′)	Amplicon Size (bp)
*ACT*	Forward: CTAAACCGCTTGTTGCTGGC	104
	Reverse: GGGGAGCTCACACGAAAGAT	
**Polysaccharides Metabolism (22)**		
*GAPA1*	Forward: TCGTTTTCATGCTTGTAACTTGT	112
	Reverse: CTTACGCCTCATTTCGCCTC	
*GAPA2*	Forward: GAAAGCGTCCTGAGCAAAGT	172
	Reverse: GCCCAGGACATACCCAAAAG	
*GAPB*	Forward: GGCAAGATGGAACTTCATGCG	106
	Reverse: ATGTGAAGTCGGGCCAAAAC	
*GAPC*	Forward: TTTTGGTCTGAGCCAGAGAGG	106
	Reverse: TGTTACCGCCTGAAAATACCT	
*BGAL5*	Forward: AGGCTCTGCTACGTTTGCTT	169
	Reverse: TCTCACGTTTCGGCTTTCGT	
*BGAL7*	Forward: AGTCTCATTGCCATTCCCCG	104
	Reverse: TGGGCGATGAATTTGGTGGA	
*GALT29A*	Forward: AGCTCTGAACGGAAAGCTCAT	186
	Reverse: GCTTGCTCACGAATACCCCA	
*GLCAT14A*	Forward:TGGTGTGACGAGGTTCAAGAGA	148
	Reverse: CAGATTCGCTGGTAACTGCCT	
*GMD1*	Forward: ATTGCTCTTGCACATCACACAC	101
	Reverse: GGCTTATAGCGGTCAACAAAAT	
*MUR1*	Forward: AGGCAAACGATTGTTGCGAG	180
	Reverse: GGATTTGTCAGCCCTTGCTT	
*MAN5*	Forward: AGCCAAGAAAATGGCGGAAT	198
	Reverse: GCGTGGATGGAATGGTGAAG	
*MSR2*	Forward: ACGAGCTTTCTCAAACAGGCA	153
	Reverse: TCGCAAGGGCTTCTAAAATGG	
*OFUT9*	Forward: GGGTTGTCCTTTGGTCTTGT	110
	Reverse: AGTTTGCGCTTGTTGTCTACC	
*OFUT20*	Forward: TTCAGGACATAGAGGAGCAGC	159
	Reverse: GTCCCCCTCCATAAAAGGCG	
*OFUT23*	Forward: GCGACTTCTTACCGGCATCT	191
	Reverse: GCCTGTCCCAAACTCTGACA	
*OFUT27*	Forward: GTTCACCGTTGCAAGACCAC	132
	Reverse: CCTTGGCTGGTGGAATGGAT	
*TPS9*	Forward:TGAGTAAGGAACAAGCCCCATC	164
	Reverse: CCTTTCCAGGCCGAGACATAA	
*TPS11*	Forward: TCCGGTCGGTGAAAGGTATG	131
	Reverse: ATCCCATCAACCACAGCCTC	
*TPPF*	Forward: TCGGGAAAACCAATGGGTGA	128
	Reverse: AGACGGCTGAACTTGAGGTG	
*TPPJ*	Forward: TACCAACTGTGCTAAGCCCT	104
	Reverse:CTGTATATTGGGTTTTGGAAGGC	
*CWINV1*	Forward: GTGACGTGTGTTTCCAGTGTG	109
	Reverse: TCAGTGTCAGCCATAAGTTGGT	
*CYFBP*	Forward: TAGTGGGCAGGGTTTAGGCA	109
	Reverse: TCGTGCGGTTAGTGTTTTACCT	
**Starch (5)**		
*At2g31390*	Forward:TGTCCGCAAACAGAAAACGTC	120
	Reverse: TGGACGCCAAAGAGGGAATG	
*DSP4*	Forward: CCCGTGTTTATCCTCGTTGGT	157
	Reverse: AAGGTGGTGGTTGACGGTG	
*NANA*	Forward: ATGCCGATCCCCAAACACA	102
	Reverse:CGAAGGTAATGCCAAATTGAGA	
*SBE2.2*	Forward:TGTCCGCAAACAGAAAACGTC	120
	Reverse: TGGACGCCAAAGAGGGAATG	
*SS2*	Forward: CGGCACAAAATCAACATGGG	104
	Reverse: CCAGGCATTCAGTTGCGAAG	
**Transport (10)**		
*STP1*	Forward: GCACTTAGCTTTGATATGCCCC	112
	Reverse: TTTAAGACCCATCGCCGTCC	
*STP5*	Forward: TCTGAGACAAACAGCCTTCC	110
	Reverse: TCCCGTGTATAAGTGCTCTACC	
*STP12*	Forward: ACGAGCTCTGCAAAGGGTTC	179
	Reverse: CTCCATCTGGTTCAACGCAC	
*STP13*	Forward: AGTGTTCGACGGGGACTCTT	146
	Reverse: ACCCCCTCTTGAGTCTTGTC	
*SWEET5*	Forward: GGGTTAGGTTGTCGTGGACT	100
	Reverse: GCTTTGTCAAGTGTGGTGCT	
*SWEET12*	Forward: TCTGACAACTACCCGCAAGC	190
	Reverse: AGGCACAGATAGTTGCCGAA	
*SWEET14*	Forward: AGCTGCCGAAAGTACCCTAC	130
	Reverse: TCGCATGTTTCTCCTTCGCT	
*SWEET15*	Forward: TGTCGCCGTTGCATTTTTGT	137
	Reverse: CTCAACTGGGTGGCCTTCAA	
*UXT2*	Forward: AGGCCTGATTGCAAGAGCTTA	148
	Reverse: CACGGGTACGTCACTCAGAT	
*UXT3*	Forward: TGCGGTTAACCTGGAAGAGG	189
	Reverse: TGTTTAGGACATCCTCCCATGC	

## Data Availability

The datasets are publicly available at NCBI, with BioSample accession: SAMN13722045 (*Nitraria roborowskii*) and SAMN13722048 (*Nitraria sibirica*), Sequence Read Archive (SRA) accession: SRR10829653 to 10829655 (*Nitraria roborowskii*) and SRR10829660 to 10829662 (*Nitraria sibirica*) (https://dataview.ncbi.nlm.nih.gov/object/PRJNA598928). (accessed on 1 February 2021)

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
