# Peer review of "Transcriptomics Reveals Host-Dependent Differences of Polysaccharides Biosynthesis in Cynomorium songaricum"

_molecules, 2021, doi:10.3390/molecules27010044_

Round 1

Reviewer 1 Report

Authors must improve the Introduction stating that why they conducted this research, what were research gaps and what aspect of the work has not been done. 

Conclusions need a strict revision, what was the core finding of the work? what is its significance and what may be done in future? 

Author Response

Reviewer 1 comments:

  1. Authors must improve the Introduction stating that why they conducted this research, what were research gaps and what aspect of the work has not been done.

Thanks for your suggestion, the statement on why we conducted this research has been added in the Introduction section: “Since C. songaricum is widely used as a traditional Chinese medicine and several pharmacological activities are largely relied on polysaccharides [4, 8]; moreover, the growth differences in C. songaricum host in the two N. roborowskii and N. sibirica have been reported [22-24], the regulation mechanism of polysaccharides biosynthesis has not been revealed.” (Page 2, lines70-74)

  1. Conclusions need a strict revision, what was the core finding of the work? what is its significance and what may be done in future?

Thanks for your suggestion, the conclusion section has been strictly revised: “From the above observations, the stem biomass and polysaccharides accumulation of C. songaricum host in N. roborowskii are significantly greater than that of N. sibirica. A total of 1,725 UR and 848 DR genes were observed in CR compared to CS, and 50 DEGs were involved in polysaccharides biosynthesis, which indicates that the polysaccharides biosynthesis in C. songaricum is host-dependent. The specific roles of candidate genes in regulating polysaccharides biosynthesis will require additional studies. ” (Page 14, line 327-332)

Reviewer 2 Report

Premise:

The manuscript by wang et al entitled "Transcriptomics reveals host-dependent differences of polysaccharides
biosynthesis in Cynomorium songaricum"  showed how host dependent influences transcriptional changes and thus differences in polysaccharides biosynthesis. Authors tested effects of transcriptional changes  and net soluble sugar content and antioxidant capacity on a per stem when c.songaricum  hosted either in Nitraria roborowskii or  in Nitraria sibirica. Authors results concluded that c.songaricum grown on Nitraria roborowskii relative to sibirica.

Overall the results provided by authors back up their claims and this results could be of interest to audiences in the field of host -parasite interaction and traditional Chinese medicine.

Comments to the authors:

  1. Authors should mention the statistical test used to obtain their Pvalue in figure 1 and 2.
  2. Number of observation used to make barplot should be mentioned in the legend of figure 2. 
  3. Authors should the quality and metrics of RNA seq data as a table. Although it is fine, I would recomment authors to transfor their table into a figure. This is not a hard work. Using simmple R plotting  will be enough. 
  4. SWEET family genes have been know to be important for sugar synthesis in various plants and accordingly they were enriched in CR compared 145 with CS. Authors should explain this briefly in their discussions.

Author Response

Reviewer 2 comments:

  1. Authors should mention the statistical test used to obtain their P value in figure 1 and 2.

According to your comments, the statistical test used in Figure 1 and 2 has been added: “A t-test was applied for independent samples, the “*” is considered significant at p < 0.05 between CR and CS.” (Pages 3, lines 88-89 and 89-100)

  1. Number of observation used to make barplot should be mentioned in the legend of figure 2. 

According to your comments, the number of observation has been added in the legend of Figure 2: “(mean ± SD, n=20)”. (Page 3, line 97-98)

  1. Authors should the quality and metrics of RNA seq data as a table. Although it is fine, I would recommend authors to transfer their table into a figure. This is not a hard work. Using simple R plotting will be enough. 

According to your comments, the tables showing the quality and metrics of RNA seq data have been transferred to Figures 3, 4, 5 and 6. (Page 3-5, lines 106-110 and 116-131)

  1. SWEET family genes have been known to be important for sugar synthesis in various plants and accordingly they were enriched in CR compared with CS. Authors should explain this briefly in their discussions.

Thanks for your suggestion, the description about the SWEET family genes has been added in the Discussion section: “SWEETs is a unique new family of sugar transporters that lead to many elusive transport steps including nectar secretion, phloem loading and post-phloem unloading as well as novel vacuolar transporters [59].” (Page 11, line 254-257)